# Putative Role of the Lung–Brain Axis in the Pathogenesis of COVID-19-Associated Respiratory Failure: A Systematic Review

**DOI:** 10.3390/biomedicines10030729

**Published:** 2022-03-21

**Authors:** Francesco Gentile, Tommaso Bocci, Silvia Coppola, Tommaso Pozzi, Leo Modafferi, Alberto Priori, Davide Chiumello

**Affiliations:** 1Clinical Neurology Unit, “Azienda Socio-Sanitaria Territoriale Santi Paolo e Carlo” and Department of Health Sciences, University of Milan, Via Antonio di Rudinì 8, 20142 Milan, Italy; francesco.gentile@unimi.it (F.G.); alberto.priori@unimi.it (A.P.); 2“Aldo Ravelli” Center for Neurotechnology and Experimental Brain Therapeutics, Department of Health Sciences, University of Milan, Via Antonio di Rudinì 8, 20142 Milan, Italy; 3Anesthesia and Intensive Care Unit, “Azienda Socio-Sanitaria Territoriale Santi Paolo e Carlo” and Department of Health Sciences, University of Milan, Via Antonio di Rudinì 9, 20142 Milan, Italy; silvia.coppola@asst-santipaolocarlo.it (S.C.); tommaso.pozzi@unimi.it (T.P.); leo.modafferi@unimi.it (L.M.); davide.chiumello@unimi.it (D.C.); 4Coordinated Research Center on Respiratory Failure, University of Milan, 20100 Milan, Italy

**Keywords:** SARS-CoV-2, COVID-19, brainstem, respiratory failure, neurological COVID, neurophysiology, neuropathology, acute respiratory distress syndrome, systematic review

## Abstract

The emergence of SARS-CoV-2 and its related disease caused by coronavirus (COVID-19) has posed a huge threat to the global population, with millions of deaths and the creation of enormous social and healthcare pressure. Several studies have shown that besides respiratory illness, other organs may be damaged as well, including the heart, kidneys, and brain. Current evidence reports a high frequency of neurological manifestations in COVID-19, with significant prognostic implications. Importantly, emerging literature is showing that the virus may spread to the central nervous system through neuronal routes, hitting the brainstem and cardiorespiratory centers, potentially exacerbating the respiratory illness. In this systematic review, we searched public databases for all available evidence and discuss current clinical and pre-clinical data on the relationship between the lung and brain during COVID-19. Acknowledging the involvement of these primordial brain areas in the pathogenesis of the disease may fuel research on the topic and allow the development of new therapeutic strategies.

## 1. Introduction

The severe acute respiratory syndrome coronavirus 2 (SARS-CoV-2) is the virus responsible of the current pandemic threat caused by the coronavirus (COVID-19). COVID-19 pneumonia manifests with fever, dry cough, hypoxia, and fatigue, requiring hospitalization in about 20% of patients [1]. Complications include acute respiratory distress syndrome (ARDS), acute myocardial injury, blood clotting abnormalities, and acute kidney injury [2,3], which are associated with dramatic morbidity and mortality [4,5].

Neurological complications may occur in COVID-19 patients and have been recently described in 30 to 80% of cases [6,7]. They include not only headache, anosmia, and myalgia, but also more severe manifestations, such as acute encephalopathy, coma, and stroke [6,7,8], as well as neuromuscular disorders, such as Guillain-Barré syndrome (GBS) [9,10]. Neurological involvement has been found more frequently in severe infections [7], and consequently, more in males than in females, and it has been associated with higher mortality in hospitalized patients [6].

Neurological involvement may stem from the potential neurotropism of SARS-CoV-2. Many areas of the central nervous system (CNS) are pathologically involved in COVID-19 patients, with the brainstem being the most common site of viral invasion and damage, suggesting it may be an important target of the virus [11]. It is the home for several neuroanatomical networks responsible for alertness, sleep/wake cycle, and cardiorespiratory regulation (Figure 1), which are richly linked to the respiratory system for breathing control and acid–base homeostasis. The presence of a brainstem dysfunction related to SARS-CoV2 infection may offer a plausible explanation for the high frequency of neurological manifestations observed in COVID-19 patients and a potential biological basis for some peculiar clinical features observed during the disease course.

Herein, we present a narrative review on the putative role of the lung–brain axis in the development of COVID-19 pneumonia and associated respiratory failure, highlighting clinical, neurophysiological, and neuropathological evidence of SARS-CoV-2 neurotropism. Furthermore, we discuss potential pathogenic mechanisms possibly contributing to the acute disease respiratory distress, as well as clinical and therapeutic implications of COVID-19-induced brainstem damage.

## 2. Methods

We carried out an extensive search on PubMed and Google Scholar databases updated until 10 January 2022. The search keywords were (“COVID-19” or “SARS-CoV-2” or “coronavirus”) AND (“respiratory failure” or “hypoxia” or “neurological manifestations” or “neurophysiology” or “neuropathology” or “brainstem” or “pathology” or “neurotropism”). Two authors (F.G. and T.B.) screened records of search outputs for pertinence to the topic and English language only. A flow chart of the systematic literature search according to PRISMA guidelines is reported in Figure 2. Titles and abstracts were then reviewed and only clinical studies with more than 50 patients were included in the analysis, while single case reports and small case series already included in cited studies were excluded, with the exception of cases which we deemed valuable to mention for the sake of the discussion. A total of 151 eligible articles were fully reviewed by the authors and further selection was made on the basis of peer-quality review and relevance to the topic, finally including a total of 97 studies in the present review. For neuropathological studies, we sought to collect relevant clinical (total number of patients, number of patients with neurological symptoms, number of patients in the intensive care unit) and morphological (neuronal loss, inflammatory, and vascular pathology) data and then summarize the studies in three groups, according to the predominant features observed (see below). Biases related to missing data or flaws in study design are appropriately discussed where needed. We also included data gathered from our own personal experience with COVID-19 patients, expanding current knowledge on the topic. Critical aspects and controversial issues were highlighted and critically discussed in an attempt to clarify current evidence and future perspectives. This systematic review has been registered to PROSPERO and received the identification number ID-316230.

## 3. Results and Discussion

### 3.1. Clinical Clues of Brainstem Involvement in COVID-19: Non-Neuro-Specific Symptoms

Smell and taste disorders are common features of COVID-19. Their frequency ranges from 5 to 98%, according to ethnicity, study design, and investigational method [12]. Studies in Europe reported a consistently high incidence of olfactory (53.7–85.6%) and gustatory (52.2–88%) dysfunction [13,14]. They present in association in over 90% of cases, suggesting a strict relationship in the development of these disturbances [13,14,15,16]. Both symptoms appear quite early and may be the first disease manifestation in about 10% of cases [13]. For this reason, many authors recommend a high index of suspicion of SARS-CoV-2 infection in front of an acute loss of smell or taste [12,17]. Interestingly, traditional signs of an upper respiratory tract infection, such as nasal obstruction and rhinorrhoea, are absent in more than half of cases with hypo-/anosmia, thus suggesting a sensorineural origin of the disorder [15,18]. Smell and taste deficits are usually transient, with a full recovery observed in about 85% within the first two months, although cases with long-term disturbances have been also reported [19]. Recognition of olfactory or gustatory disorders is relevant due to its prognostic implications, as they have been associated with mild disease and reduced in-hospital death [6,20], and they tend to be absent in more severe cases [14].

Another peculiar feature of COVID-19 is the low frequency of dyspnoea among patients with radiologic evidence of pneumonia and hypoxia, referred to as “silent” or “happy hypoxia” or “non-dyspnogenic acute hypoxemic respiratory failure” [21,22,23]. Dyspnoea is the conscious perception of the difficulty in breathing which, besides hypoxia, may be stimulated by hypercapnia and inflammation of the airways/parenchyma. In a cohort of 1099 patients, dyspnoea was reported in only 19%, rising to no more than 38–55% in those patients who required supportive ventilation [2,24]. In comparison, frequencies of dyspnoea were significantly higher in similar respiratory illnesses, such as Middle East respiratory syndrome (MERS)-CoV (72%), influenza virus (82%), community-acquired pneumonia (94%), and respiratory syncytial virus (95%) [25,26,27,28]. Nonetheless, other clinical signs of respiratory failure, such as tachycardia and tachypnoea, which represent autonomic neural responses to hypoxia, are commonly observed in COVID-19 patients. This dissociation between the autonomic response to respiratory failure and its conscious perception led some authors to suggest a disorder of interoception, in which the transmission of neural signals from the lung to the CNS is altered [29]. Strikingly, dyspnoea rates were similar between SARS-CoV-2 and SARS-CoV-1 viruses, for whom neurotropism has been clearly established [28,30].

### 3.2. Neuro-Specific Symptoms

The rich network of connecting fibers and nuclei in the brainstem may predispose to a wide variety of symptoms, ranging from non-specific to more florid and localized manifestations. Altered mental status and acute encephalopathy are present in hospitalized COVID-19 patients, ranging from 31 to 49% in large, multicentre cohort studies [6,8]. Acute encephalopathy is common in the setting of a severe infection and shows frequency rates of up to 70% in the intensive care unit (ICU), usually related to sepsis, hypoxia, pharmacologic sedation, toxic exposure, or metabolic dysfunction [31]. Many of these factors are detected in COVID-19 patients during an ICU stay, potentially explaining the high rate of *delirium* observed. Nonetheless, one study investigated the relationship between clinical, radiologic, EEG, and laboratory findings in COVID-19 cases with neurologic manifestations [32]. Despite the fact that a high number of encephalopathic patients resulted from common biological abnormalities, such as liver/renal failure and electrolyte disturbances, a small group of patients had no identifiable cause of brain injury and were labeled as COVID-19-related encephalopathy (CORE), which was significantly associated with clinical signs of frontal lobe and brainstem impairment [32].

Seizures or seizure-like events are uncommon in COVID-19 patients, reported in about 1% of the affected general population, raising up to 10% in restricted cohorts with neurological features [6]. Clinical manifestations range from focal involuntary movements to generalized tonic-clonic seizures [32,33,34]. Myoclonus has been reported in some studies as the most frequent abnormal involuntary movement observed in COVID-19 patients. Nonetheless, some peculiar features were described, with a preferred generalized or multifocal distribution, simultaneously involving the face, trunk, and extremities; jerks were either spontaneous or triggered by voluntary and sensory stimuli, sometimes combined with opsoclonus and ataxia, possibly suggesting a subcortical origin of the myoclonus [35,36]. Most of these patients were admitted to the ICU, where comorbidities and complications, such as metabolic derangements, hypoxia, and medications, may explain the occurrence of myoclonic jerks. However, myoclonic jerks were described more frequently in severe COVID-19 compared to other viral illnesses [37], as well as being reported in mild/moderate COVID-19, even in the absence of alternative explanations such as hypoxia [38]. Interestingly, jerks appeared days after the onset of the respiratory symptoms and may last up to several weeks [36,39]. Response to treatment, which included anti-epileptic drugs and immunotherapy, was also variable [36].

Careful evaluation of brainstem reflexes may allow a better definition of the neuroanatomical substrates involved in SARS-CoV-2 infection. Two studies performed in severe COVID-19 patients revealed that the glabellar, corneal, oculocephalic, and cough reflexes, whose integrity is related to intrinsic brainstem neurocircuits, are absent or severely impaired [40,41]. Interestingly, in one of these studies, brainstem damage manifested with failure of the central respiratory drive, leading to prolonged ICU stay and mechanical ventilation despite resolution of the pneumonia [41].

Finally, it is worth noting that immune-mediated damage of the brainstem has been rarely reported [42], as well as cerebrovascular accidents affecting the posterior cranial fossa [43,44].

### 3.3. Neurophysiological Correlates

Electroencephalography (EEG) showed non-specific features in the majority of cases, such as a diffuse, symmetrical background slowing, a generalized rhythmic delta activity, or generalized periodic discharges with triphasic morphology [45,46]. Similar findings were observed in CORE patients, where EEG showed frontal-predominant periodic discharges [32]. Of note, one study reported an alpha coma EEG pattern in 5 out of 19 severe COVID-19 cases, further supporting the hypothesis of brainstem dysfunction in the disease [47]. However, this study should be interpreted with caution, as the lack of details about clinical-EEG correlations and the absence of confirmatory studies do not allow to generalize these findings.

Despite most studies not reporting localizing features in EEG, studies on myoclonus in COVID-19 may help to shed light on viral neurotropism and damage. EEG recordings during myoclonic jerks failed to reveal cortical neurophysiological correlates with the abnormal movements [36,37,38,48]. Strikingly, we observed myoclonus and encephalopathy in four patients with COVID-19, who did not suffer respiratory failure or major organ damage. Myoclonia were waxing and waning, bilaterally distributed, involving limbs, trunk and the head, without a distal-to-proximal gradient of appearance. EEG showed the presence of generalized/lateralized periodic discharges, predominating in the anterior regions, in a diffuse slowing of the background activity (Figure 3A; unpublished data). Co-registration of EEG with poly-electromyography showed that the electrophysiological abnormalities were also temporally unrelated to the myoclonic jerks (Figure 3B; unpublished data), strongly supporting a subcortical origin. Another study showed similar findings in a COVID-19 patient with multifocal myoclonus, without any evidence of EEG phase-locked discharges [49]. Interestingly, a brainstem localization of the myoclonus has been further suggested by Newcombe and co-workers, who demonstrated reduced diffusivity on diffusion tensor imaging MRI in the ponto-mesencephalic region of a small series of COVID-19 patients [50]. Post mortem examination in one of them revealed marked inflammation in the dorsal medulla, with widespread microglial activation and nodules.

Neurophysiologic assessment of the brainstem function may be of value in these patients. The blink reflex appeared abnormal in all 11 cases tested in one study [40] and one out of four in another [51], supporting potential damage in the ponto-medullary centers of the brainstem (Figure 3C). Other modalities of neurologic brainstem assessment, such as somatosensory evoked potentials and brainstem auditory evoked potentials, have been performed but were unremarkable [51,52]. The contradictory findings observed across different studies may be related to the different neurophysiological methods, as they assess slight but significantly different circuits in the brainstem. The preferential impairment of some networks over others suggests that brainstem dysfunction may be caused by selective neurotropic damage of SARS-CoV-2 rather than a diffuse injury caused by systemic inflammation and multi-organ failure, as instead supported by some neuroimaging studies [53,54].

### 3.4. Neuroimaging Studies

Neuroradiological findings supporting a preferential brainstem or medullary involvement are rare, mainly comprising single case reports or short series [55]. This limitation is probably due to the severity of the respiratory distress underlying the acute phase of the disease, limiting the access to second-line diagnostic tools, such as magnetic resonance imaging (MRI) and positron emission tomography (PET). Some authors have recently suggested that SARS-CoV-2 infection may trigger encephalitis with prominent Parkinsonism and distinctive brain metabolic changes in the brainstem, mesial temporal lobes, and basal ganglia [55]. These data fit with other studies describing Parkinsonian syndromes developing soon after COVID-19 [56,57,58]. Some of these cases closely resemble the so-called “encephalitis lethargica”, a neurological syndrome that spread in the period 1916 to 1930 [59], but patients without encephalitis have been described as well [58].

Moreover, other studies have recently identified a specific profile of brain PET hypometabolism in long COVID patients, also including the bilateral pons and medulla [60,61].

Further studies are needed to assess whether these findings are coincidental or not, possibly identifying delayed syndromic correlates.

### 3.5. Pathogenic Mechanisms

SARS-CoV-2 gains access to host cells by interacting with the angiotensin-converting enzyme 2 (ACE2) and the transmembrane protease serine protease 2 (TMPRSS2), which are vital for viral replication and host invasion. A severe inflammatory response and hypercoagulability are highly responsible for the clinical manifestations of the disease, leading to widespread lung damage along with a myocardial and renal injury.

Neuropathologic evidence of SARS-CoV-2 related lesions recently emerged, with both inflammatory and vasculopathic features frequently coexisting upon microscopic examination (Table 1) [11,62,63,64,65,66,67,68,69,70,71,72,73,74,75,76,77,78,79].

Signs of innate neuroinflammation, including microglial and astrocytic activation, are prominent in the brains of COVID-19 patients, demonstrating a preferential distribution into the brainstem, followed by the olfactory bulb (Table 1). Microglial nodules and neuronophagia, although less frequently reported, have been appreciated in the same sites and may be an expression of local viral infection, as frequently encountered in encephalites of viral and autoimmune etiology [11]. These signs have been particularly observed along the pontomedullary junction of the brainstem, including the dorsal motor nucleus of the vagus nerve, the nucleus ambiguous, the solitary tract/nucleus, and pre-Bötzinger complex [67,75], key structures for the control of respiration. These neuroanatomical substrates may underlie the respiratory dyssynergia observed in a significant amount of COVID-19 ventilated patients, with some studies reporting that patient–ventilator asynchronies may occur in up to 5% of respiratory acts registered during long-term monitoring of COVID-19 patients, potentially contributing to worsening the lung damage and associated mortality [80]. Despite a similar frequency [73], the brainstem-predominant microgliosis appears as a unique feature of COVID-19 compared to the diffuse CNS distribution observed in septic patients [69]. Strikingly, brainstem microgliosis represented the only neuropathological finding in COVID-19 patients with mild/moderate disease and in those who were neurologically asymptomatic, supporting its link with SARS-CoV-2 infection.

Signs of microvascular lesions are common and include hypoxic, ischemic, and hemorrhagic lesions, microscopically detected in several brain areas (Table 1). Neuroimaging studies further support a vascular etiology for some of the neurological manifestations of COVID-19, such as acute ischemic infarcts, micro-/macrohemorrhages, posterior reversible encephalopathy syndrome (PRES), hypoxic-ischemic encephalopathy, acute necrotizing encephalopathy, and non-specific leukoencephalopathy [81,82,83]. Many factors may contribute to the development of COVID-19 cerebral vasculopathy. Systemic features of the disease, such as the “cytokine storm” and the hypercoagulability, may cause diffuse endothelial dysfunction and thrombosis, exacerbated during the stay in ICUs due to the respiratory failure, septic complications, end-organ failure, and invasive life support, including mechanical ventilation and extracorporeal membrane oxygenation. Nonetheless, a study has recently shown that in experimental models and humans with SARS-CoV-2 infection, the virus successfully invades the CNS, causing metabolic alterations in infected neurons, and exacerbates hypoxemic effects on nearby neuronal populations, predisposing to significant neurovascular damage [79]. Furthermore, some studies have reported viral detection in the endothelial cells of infarcted areas [62,66,68,69], suggesting that direct viral effects may contribute to cerebral endothelial dysfunction.

The pathological detection of SARS-CoV-2 proteins, a marker of CNS infection, in neurons and glial cells has been reported in some studies, found predominantly in the olfactory bulb, the brainstem, and V, IX, and X cranial nerves (Table 1). The focal distribution of SARS-CoV-2 and the overlapping microglial activation suggest an active infectious process, which may result in local injury with neuronal and axonal loss, as witnessed in some human and experimental studies [62,73,76]. Interestingly, Matschke and co-workers found that disease duration inversely correlated with the CNS viral load [11]. It should be noted that viral detection was not necessarily accompanied by lymphocytic infiltration. Indeed, some authors postulated that SARS-CoV-2 may evade the immune response by dampening interferon response [84,85,86,87], allowing successful infection and dissemination in the host.

The virus appears to target selective neuroanatomical structures, such as the reticular formation, the vagal nuclei, the solitary tract/nucleus, and ventral respiratory column, which are essential in the neural control of respiration and other neurovegetative functions [88]. Dysfunction of these circuits may impair wakefulness, autonomic functions, and spontaneous respiration, as commonly reported in COVID-19 patients (see above). Furthermore, the spatial patterns of viral neuroinvasion point towards neuronal transport as the main portal of entry to the CNS. The olfactory, trigeminal, glossopharyngeal, and vagus nerves innervate the whole respiratory tract, where a high viral load is present during the acute phase of the disease. The virus may gain access to the peripheral nerve endings and then travel along the axonal route in a retrograde fashion. The identification of SARS-CoV-2 along the cranial nerves and their central nuclei corroborates this hypothesis [62]. Furthermore, the virus has also been localized in the carotid body [89], suggesting local dysfunction of the oxygen-sensing system and representing a potential route of entry through the glossopharyngeal nerve. However, it should be noted that, besides the olfactory nerve [68,90], evidence of active viral migration along the peripheral nerve routes is still lacking. In addition, studies about the expression of the ACE2 and TMPRSS2 receptors in the central nervous system reported contradictory results [11,68,78,79], highlighting important gaps in the knowledge of the underlying mechanisms of neuronal infection.

### 3.6. Clinical and Therapeutic Implications

Impaired consciousness after cessation of sedation may occur for both neurological and systemic reasons. A CT scan is essential to exclude acute intracerebral hemorrhage which may increase intracranial pressure. Eventually, a brain magnetic resonance (MRI) may reveal critical illness-associated microbleeds, a common complication in the ICU setting which may impair consciousness [91]. Systemic causes, such as hypoxia, metabolic, and electrolytes alterations, and organ failure, should also be accounted for in case of suspected encephalopathy [31,92,93].

Failure of weaning from ventilation may also be caused by dysfunction in the peripheral nervous system. Acquired generalized weakness and weaning failure may be caused by disorders such as GBS and critical illness polyneuropathy/myopathy (CIP/CIM), which have been described in COVID-19 patients. Population studies reported an increased incidence of GBS across individuals infected with SARS-CoV-2 [10,94], including rare variants such as Miller Fisher syndrome, polyneuritis cranialis, facial diplegia, and pharyngo–cervical–brachial forms [9,95,96]. The incidence of CIP/CIM is high in severe COVID-19 patients, reaching about 10% in prospective studies [97,98]. The clinical impact of these disorders is relevant, as they may lead to thromboembolic events related to immobilization, prolonged mechanical ventilation, and ICU stays, raising the risk of in-hospital death [10,97,99]. The request of neurophysiological studies is thus of utmost importance for the identification of these disorders. The involvement of cranial nerves, the presence of autonomic dysfunction, as well as the pattern of electrodiagnostic findings, may help in the distinction between GBS and CIP/CIM, essential for the rapid institution of immunotherapy in case of GBS.

Dysfunction along these brainstem circuits may contribute to exacerbating critical illness among patients admitted to the ICU, with significant prognostic implications [92]. The combination of direct SARS-CoV-2 infection as well as concomitant hypoxic, infectious, metabolic, and vascular insults, likely contributes to significant neuronal damage and neuroinflammation in these areas. Brainstem dysfunction may explain the prolonged hospital stay in ICU-admitted COVID-19 patients, which may be prolonged for several weeks [3,100]. Several studies in non-COVID-19 patients showed that the absence of cough and/or gag reflexes, signs of lower brainstem dysfunction, are predictive for ARDS development [101]. Furthermore, the assessment in brainstem responses, such as cough and oculocephalic reflexes, may predict 28-day mortality and difficulty awakening after sedation withdrawal in critically ill cases, respectively [102]. Finally, disruption in breathing neural control may cause respiratory asynchrony and irregular patterns, limiting efficacious gas exchange and making adaptation to mechanical ventilation a tough task.

Several studies increasingly recognized the existence of a post-viral syndrome, also known as long COVID-19, which may persist for long periods after the resolution of the infection. Chronic headache, myalgias, fatigue, cough, and cognitive impairment have been reported in 30–80% of COVID-19 survivors, persisting for about 1 to 6 months, with a significant impact on the quality of life [103,104,105].

In a 3-month follow-up study, 55% of patients still reported neurological symptoms, such as fatigue, myalgia, mood changes, memory loss, and mild sensory-motor deficits. MRI scans performed in these patients revealed significant changes in grey matter volumes of several brain areas as well as diffuse damage in the white matter, proving neuroanatomical substrates at the base of the long-term consequences of SARS-CoV-2 infection [106].

Strikingly, silent hypoxia may also persist after recovery from the acute disease. One study showed that as much as 50% of COVID-19 patients may show asymptomatic exercise-induced hypoxia upon discharge [107]. Raising the threshold of dyspnoea sensation may favor the development of silent heart and brain ischemic events in the long term, which are known predictors of morbidity and mortality [108].

Some authors have recently suggested that long-COVID may be caused by SARS-CoV-2 tropism to the brainstem [109]. The occurrence of cardiorespiratory, gastrointestinal, and neurological symptoms may be linked to the neurovegetative functions exerted by the numerous brainstem nuclei. Previous studies have shown that microglial activation may persist for several weeks after an acute infectious insult [110], with the brainstem being particularly susceptible to sustained neuronal damage induced by systemic illness [111]. These findings provide rationale support for the involvement of this region also in the long-COVID-19 syndrome.

Therapeutic interventions after recognition of brainstem damage are scarce. No pharmacologic treatment has been tested for neuroprotection in COVID-19, as most research concentrated on the development of effective drugs to dampen viral infection as well as the maladaptive inflammatory response. Still, some authors advocate for the role of non-pharmacologic interventions, such as non-invasive vagal nerve stimulation, in COVID-19 patients [112]. Experimental studies demonstrated that this approach may reduce systemic inflammation and aids in recovery in models of sepsis [113,114]. To date, only case reports have been described, which showed the safety of the intervention as well as the potential contribution to the clinical recovery of the tested patients [115,116]. Nonetheless, current evidence is insufficient to suggest a significant efficacy of vagal nerve stimulation in these patients. In this sense, two clinical trials (NCT04382391and NCT04368156) are ongoing to address the issue.

## 4. Conclusions

Several lines of evidence support a role of brainstem dysfunction in COVID-19, potentially affecting disease course and recovery from respiratory failure. Neurotropism of SARS-CoV-2, with consequent activation of the innate neuroinflammatory response, has been observed in autoptic studies, with preferential involvement of the olfactory bulb and the pontomedullary region, site of the cardiorespiratory centers. Clinical and neurophysiological signs of brainstem dysfunction have been reported, which may have important implications in terms of disease severity, hospital stay, and survival. Furthermore, prolonged damage may affect the recovery of COVID-19 patients, leading to persistent symptoms and low quality of life. Further investigations about the role of the brainstem in COVID-19 are needed to improve diagnostic assessment and prompt research for new therapeutic strategies.

## Figures and Tables

**Figure 1 biomedicines-10-00729-f001:**
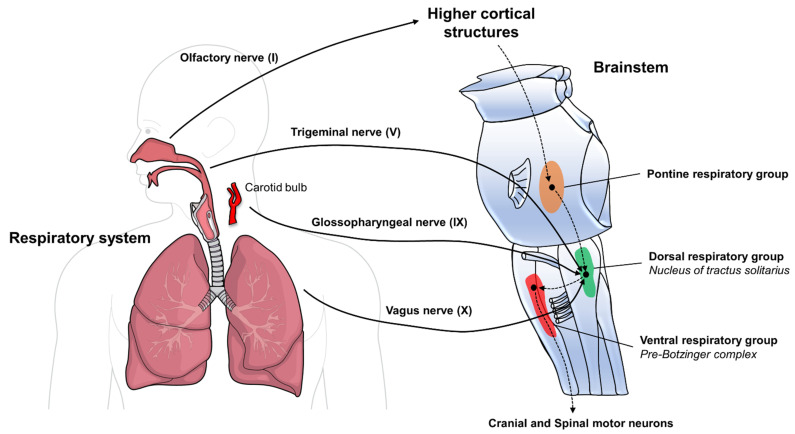
Anatomy of the lung–brain axis. Sensory inputs from the respiratory tract convey to the central nervous system through cranial nerves, delivering information about special sensation from the nose (olfactory (I) nerve) and somatic sensation from the upper respiratory mucosa (trigeminal (V) nerve), large airways (glossopharyngeal (IX) nerve), and the lungs (vagus (X) nerve). In addition, the IX nerve also transports inputs from the carotid bulb, essential for gas exchange and breathing regulation. Afferent signals converge on the nucleus of the tractus solitarius (NTS) in the pontomedullary region of the brainstem, allowing tight monitoring of the respiratory function and surveillance on potential noxious stimuli. In the NTS, some neuronal populations belong to the dorsal respiratory group (DRG), which receives information from peripheral chemoreceptors about gases’ status, lung mechanisms, and tissue damage. Further modulation of the DRG function comes from the higher cortical structures through the pontine respiratory group (PRG) station. The DRG then conveys signals to the ventral respiratory group (VRG), including the preBötzinger complex, which, through efferent connections to cranial and spinal motoneurons, is responsible for the spontaneous rhythmic pattern of respiration.

**Figure 2 biomedicines-10-00729-f002:**
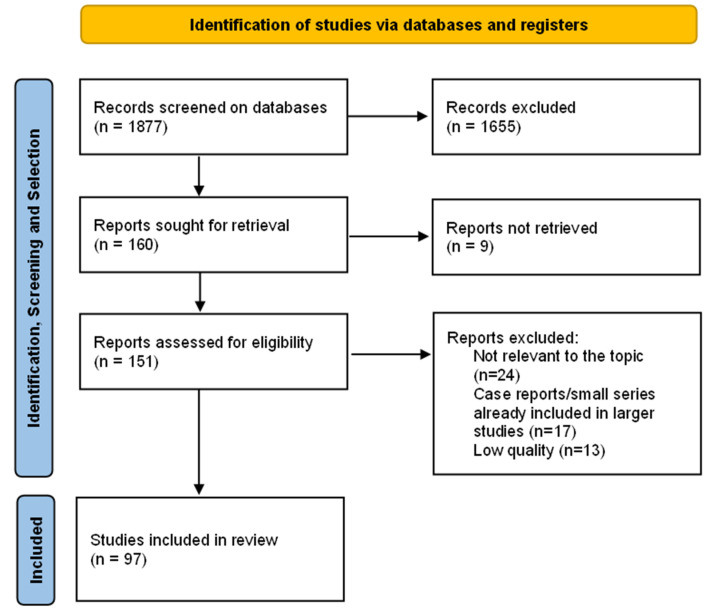
Flow diagram of the literature research, screening, selection and final inclusion in the final analysis.

**Figure 3 biomedicines-10-00729-f003:**
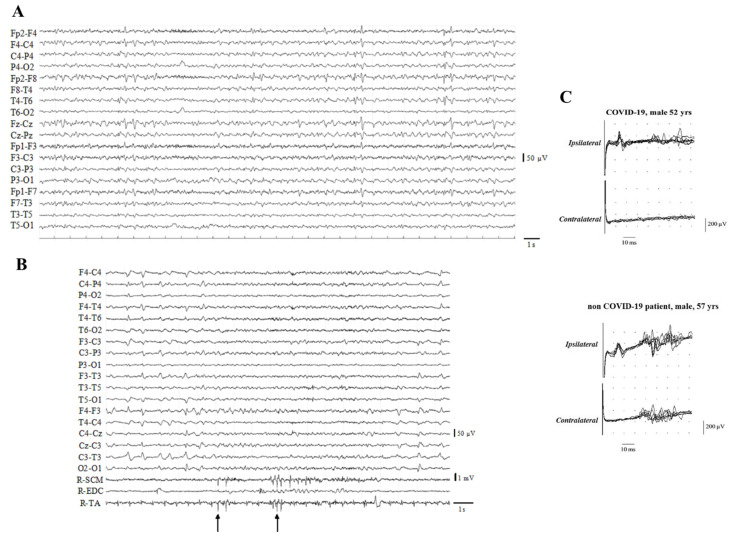
Neurophysiological findings in our series of COVID-19 patients with myoclonus. (**A**): Generalized periodic discharges, with a right hemisphere predominance, mainly recorded at the parasagittal and midline regions (unpublished data). (**B**): black arrows indicate waxing and waning myoclonic jerks. Note that these movements do not temporally correlate with periodic lateralized discharges at EEG, without a prominent proximal-to-distal gradient of appearance; all these features suggest a sub-cortical origin of the myoclonus (surface poly-EMG recorded from the right *sternocleidomastoid*, *extensor carpi radialis longus*, and *tibialis anterior* muscles; unpublished data). (**C**): Blink Reflex (eight superimposed traces) recorded in a COVID-19 patient (top) and a non-COVID-19 patient (bottom). In the former, ipsilateral RII responses had markedly prolonged latencies and contralateral RII were absent, suggesting a pontomedullary lesion (modified with permission from [40]).

**Table 1 biomedicines-10-00729-t001:** Patterns of neuropathological findings in COVID-19.

Pathology Pattern	N	NeS	N (%) ICU	Vascular Damage				Inflammatory Response			SARS-CoV-2 Detection	References
				Hy	I	T	Hem	BI	MA	MN/N		
*Inflammatory*	99	5/5	33 (33)	+/++	-/+	-	-	+/++	+++ (>BT)	++/+ (>BT)	++ (CN V, IX, V, BT, OB)	[11,62,69,73,74,76,77]
*Vascular*	94	24/39 (61)	18/29 (62)	+/++	+/++	++	+/++	+	++ (BT)/NR	-/NR	-/+	[64,65,66,71,77,79]
*Inflammatory & Vascular*	120	23/83 (28)	68/120 (57)	++	+/++	+/++	+	+/++	+++ (>BT, OB)	++/+ (BT)	++ (CN V, BT, OB)	[63,67,68,70,72,75]

Legend: +, mild; ++, moderate; +++, severe. NeS, neurological symptoms; n (%) ICU, N patients admitted in ICU; BT, brainstem; CN, cranial nerves; NR, not reported; OB, olfactory bulb. Neuro sympt, neurological symptoms N (%); Hy, hypoxic lesions; I, infarcts; T, thrombi; Hem, hemorrhages; BI, brain inflammation; MA, microglial activation; MN/N, microglial nodules/neuronophagia.

## Data Availability

Not applicable.

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
