# Peer review of "Putative Role of the Lung–Brain Axis in the Pathogenesis of COVID-19-Associated Respiratory Failure: A Systematic Review"

_biomedicines, 2022, doi:10.3390/biomedicines10030729_

Round 1

Reviewer 1 Report

This is a very well written manuscript about a highly important topic. However, I have two comments and suggestions which may improve the quality of this paper. First, can you please comment on sex differences of acute respiratory syndrome coronavirus. Are men more affected than women?

Secondly, are there any neuroimaging studies which indicate dysfunction of the brainstem? Please comments and discuss. 

Author Response

We thank very much the Reviewer for her/his valuable and useful comments.

In the revised version of our manuscript, we answered all the points raised and the paper improved.

In particular, we have added a brief comment about sex differences of acute respiratory syndrome and related neurological complications (line 42).

Moreover, we have added a whole paragraph discussing neuroradiological evidence supporting a preferential brainstem involvement (lines 217-234) and references have been updated accordingly. 

Although neuroradiological findings are rare, mainly comprising single case reports or short series, some Authors have recently suggested that SARS-CoV-2 infection may trigger an encephalitis with prominent parkinsonism and distinctive brain metabolic changes in the brainstem, mesial temporal lobes and basal ganglia. These data fit with other studies describing parkinsonian syndromes developing soon after COVID-19. Finally, other studies have recently identified a specific profile of brain PET hypometabolism in long COVID patients, also including the bilateral pons and medulla.

Reviewer 2 Report

In this review, authors are compiling the published evidence, clinical or pre-clinical data on the relationship between brain and lung during COVID-19.

Comments:

  • Adding a glossary of words would be a good addition to make the review more clear and comprehensive.
  • Line 37,38: rephrasing required. This whole paragraph rephrasing needed.
  • Line 107: claiming “permanent disturbance” is far stretch, may be using long term disturbance is better as the data is just for 1-2 years yet.
  • Fig 1 needs improvement. It’s hard to understand the arrow directions of cranial nerves. Also, in the brain stem part of the figure, figure legend doesn’t match with the region specified on the figure like pontine respiratory group, vagus nerve etc. Need to make it consistent.
  • Fig 2: resolution needs to be improved and also would be nice to add the description in the figure itself like what the arrows indicate.
  • Table 1 is too cramped with lot of information. Would be nice to break down the information more in the table.

Author Response

We thank the Reviewer for her/his valuable and very useful comments. In the revised version of our manuscript, we have addressed all the points raised and the manuscript improved.

  • Lines-37-41. We have totally re-phrased the entire paragraph.
  • Line 109. According to the the Reviewer, the term "long-term disturbances" has been introduced.
  • The Figure 1 has been improved.
  • Legend has been modified in a more readable way and the Figure has been also re-submitted as a separate file, in a high-resolution format.
  • Both Legend and footnotes have been modified.
  • Overall, English language and style have been reviewed and improved, where necessary.